# Object Tracking for an Autonomous Unmanned Surface Vehicle

**Min-Fan Ricky Lee** [1,2,*] and **Chin-Yi Lin** [1]

1   Graduate Institute of Automation and Control, National Taiwan University of Science and Technology, Taipei 106335, Taiwan; m10812013@mail.ntust.edu.tw
2   Center for Cyber-Physical System Innovation, National Taiwan University of Science and Technology, Taipei 106335, Taiwan
*   Correspondence: rickylee@mail.ntust.edu.tw

**Abstract:** The conventional algorithm used for target recognition and tracking suffers from the uncertainties of the environment, robot/sensors and object, such as variations in illumination and viewpoint, occlusion and seasonal change, etc. This paper proposes a deep-learning based surveillance and reconnaissance system for unmanned surface vehicles by adopting the Siamese network as the main neural network architecture to achieve target tracking. It aims to detect and track suspicious targets. The proposed system perceives the surrounding environment and avoids obstacles while tracking. The proposed system is evaluated with accuracy, precision, recall, P-R curve, and F1 score. The empirical results showed a robust target tracking for the unmanned surface vehicles. The proposed approach contributes to the intelligent management and control required by today's ships, and also provides a new tracking network architecture for the unmanned surface vehicles.

**Keywords:** unmanned surface vehicle; artificial intelligence; deep learning; object tracking; surface robot

## 1. Introduction

In recent years, due to experimental and communication difficulties, unmanned surface vehicles (USVs) have fallen far behind unmanned aerial vehicles (UAVs) and unmanned ground vehicles in the field of artificial intelligence (AI) research and development. However, USV not only has good future development, but also has a wide range of applications. In the military, it can be used for waypoint patrols [1,2], gathering intelligence, surveillance, and reconnaissance [3,4]. For civilian purposes, it can be used to assist in finding people who fall into the water [5], testing water quality [6,7], and so on. In the treacherous and ever-changing marine environment, collision avoidance and target tracking are the prerequisites for USV to perform tasks, and therefore become the key development direction in the USV research field [8]. The problems encountered in target tracking can be roughly divided into four categories: the shape change of the target, the scale change of the target, the occlusion and disappearance of the target, and the blurred image [9]. Target tracking methods are mainly divided into two categories, one is a filter algorithm, and the other is a deep learning algorithm. Using the Particle Filter method, based on particle distribution statistics [10]. First, the tracking target is modeled, and a similarity metric is defined to determine the degree of matching between the particles and the target. In the process of target search, it will sprinkle some particles according to a certain distribution (such as uniform distribution or Gaussian distribution), count the similarity of these particles, and determine the possible position of the target. Although this method is fast, it easily reduces the accuracy and stability of tracking. Using deep learning methods, and using deep learning to train the network model, the obtained convolution feature output performance is applied to the correlation filtering tracking framework, to obtain better tracking results [11]. This method obtains better eigenvalues, and improves the accuracy and stability of tracking, but at the same, time it also brings an increase in the amount of calculation.

Due to the floating and undercurrents of the water, many USV control systems have been proposed. Among the non-AI methods, [12] proposed the use of proportional-integral-derivative controller to control the motion of the USV. By establishing the three-degree-of-freedom kinematics and dynamics model of the USV, the heading tracking controller was designed based on the output feedback control method. In addition, [13] also proposed a sliding mode control method based on Kalman filter for the heading control of the water jet propelled USV in the horizontal plane. With Nomoto model, a heading controller based on sliding mode control is designed and the Sigmoid function is introduced to improve the traditional exponential approximation rate.

In the AI method, ref. [14] proposed an alternative navigation system when there is no global positioning system (GPS). USV uses the simultaneous localization and mapping framework to perform relative navigation with respect to the surrounding coastline and uses B-spline to parameterize coastline features for effective map management. A planning algorithm based on deep reinforcement learning was proposed to find the shortest collision avoidance path for USVs [15]. Solutions in target tracking against various certainties (such as light changes, different scenes, or occlusion of targets) have been proposed according to the surveyed literature.

A TensorFlow framework for moving object detection was proposed [16]. The proposed method is based on convolutional neural networks (CNN) target tracking algorithm for robust target detection. The dynamic frame rate optimization and selection of adaptive parameters according to the scene and content of the input video were proposed [17]. A general algorithm was proposed to estimate the perspective image area occluded by the object [18]. By connecting the real environment and the perspective space, the two coordinate spaces create a flexible object tracking environment.

When these algorithms are applied to vehicles, they face more challenges. An object tracking algorithm for UAVs using robust multi-collaborative tracker is proposed [19], which can provide additional object information and modify the short-term tracking model in time. In short-range maritime surveillance, X-band maritime radar is used to capture objects in an extended area with different intensities [20]. Combining the position, shape, and appearance of the target, multiple kernel correlation filters are proposed to track a single target in a real marine radar.

According to the above-surveyed literatures, the tracking methods developed today have insurmountable problems, such as the target of Kernelized Correlation Filter cannot be recovered after being completely occluded [21], there are many false positives in the target tracking of Tracking-Learning-Detection Tracker [22], and the Median Flow Tracker will fail in the case of moving drastically [23]. However, most of the current deep learning research has only a single neural network as the framework of artificial intelligence, and the use of a single neural network can easily cause target tracking and identification failure. This paper uses two parallel neural networks to share feature parameters to discuss improving the accuracy and recall of target tracking. In addition, in this paper, the use of self-made USVs combined with software systems were used to improve the current target tracking systems of USVs, which are mostly the shortcomings of traditional methods. This research predicts that the proposed Siamese architecture can improve the common target tracking problems in the previously proposed methods. Among them, the stability and accuracy of the system will be verified through changes in light, changes in viewpoint, and changes in obstructions. In terms of application to USV, this paper will use two different waters for testing: a large and bright swimming pool, and a narrow pond with a complex environment. Since the USV produced in this study is a catamaran, it is expected to overcome the environmental gap.

To meet the above requirements, the system will have three modes. The first mode is a 360-degree fixed-way cruise. The 360-degree platform of the camera mounted on this USV will make a circle in 60 s and take a photo every five seconds. Therefore, it will be used to stitch the panorama to ensure the entire domain is being monitored by the system. The second mode is to search for a suspicious target, grab the center of mass through the feature frame extracted by the algorithm, and calculate the gap to control the motor to feedback

and follow the target. The third mode is to maintain a safe distance from the target and follow at a constant speed. After a while, the buzzer will give a warning and emit a laser light to warn the object. This research can be said to create a new hardware design and algorithm development for the USV field.

## 2. Materials and Methods

Deep learning is applied to improve the conventional object recognition and tracking for an autonomous USV. The hierarchical control architecture is shown in Figure 1. It can be divided into high-level control and low-level control. The environmental variables will be transmitted to the robot through wire or wireless via their respective sensors in the remote section. However, the wireless signal on the water is not stable enough for the AI model to deploy remotely to control the USV via wireless communication. The centralized architecture is adopted in this paper (wire communication). The remote AI is placed onboard the USV as edge computing to avoid abnormal communication between local and remote.

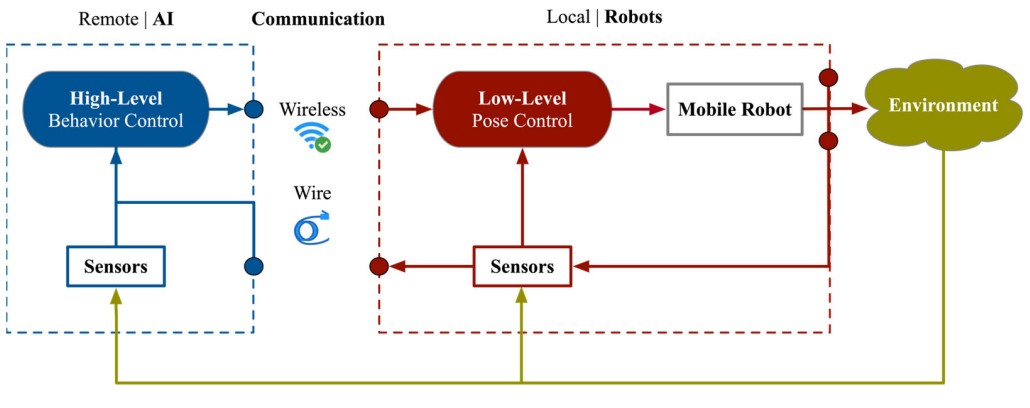

**Figure 1.** The hierarchical control architecture diagram.

The high-level control of behavior is shown in Figure 2. The information obtained by the sensor can be planned by reasoning first, or it can be divided into four behaviors: goal-seeking, obstacle avoidance, trajectory tracking, and formation keeping. Perform low-level robot behavior control. The task of formation keeping is to maintain a constant pose between the USV and target for object recognition and tracking.

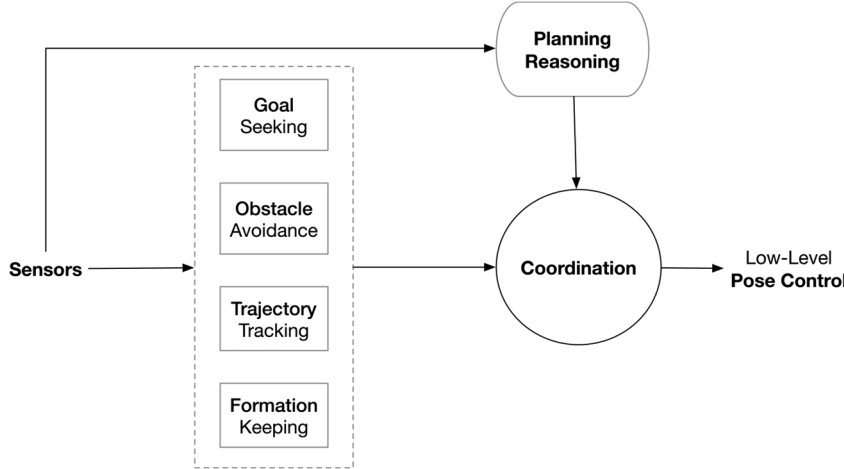

**Figure 2.** Architecture of high-level control vehicle.

## 2.1. Robot System

According to previous studies, there is a huge difference between monohull and catamaran used for USV [24–26]. Due to the design of the wave-piercing body, the catamaran has the advantages of being relatively stable, lighter overall, high capacity, and not easy to capsize. The hulls on both sides are also relatively slender, and the resistance of the water flow is reduced during sailing, so the speed is increased, and the accuracy of target tracking can be greatly improved. This paper refers to the control method proposed by Qiang Zhu [27]. The hull adopts a catamaran design and uses twin propellers as the power source. Since the USV is manufactured by the author, the rest of the parts will use the Fusion 360 integrated 3-dimensions (3D) design. The tail of the fuselage is equipped with two sets of parallel power propellers, which can achieve movement by adjusting the running speed and direction of the twin propellers. The USV hull design is shown in Figure 3. The power components of this USV are placed at the bottom of the aft end of the hull, so the motor will be submerged underwater.

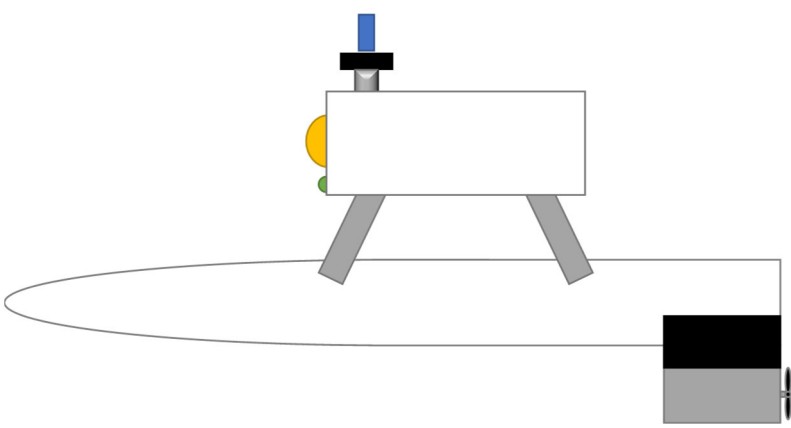

**Figure 3.** USV envisioned design drawing.

## 2.2. Algorithm

The target tracking and response system based on deep learning is applied to USV. In order to avoid image loss, cruise and image stitching are used to improve the accuracy of target tracking, and a new Siamese network method is adopted. After the system starts, it will automatically navigate to search and detect the most panoramic view. The overall system diagram is shown in Figure 4.

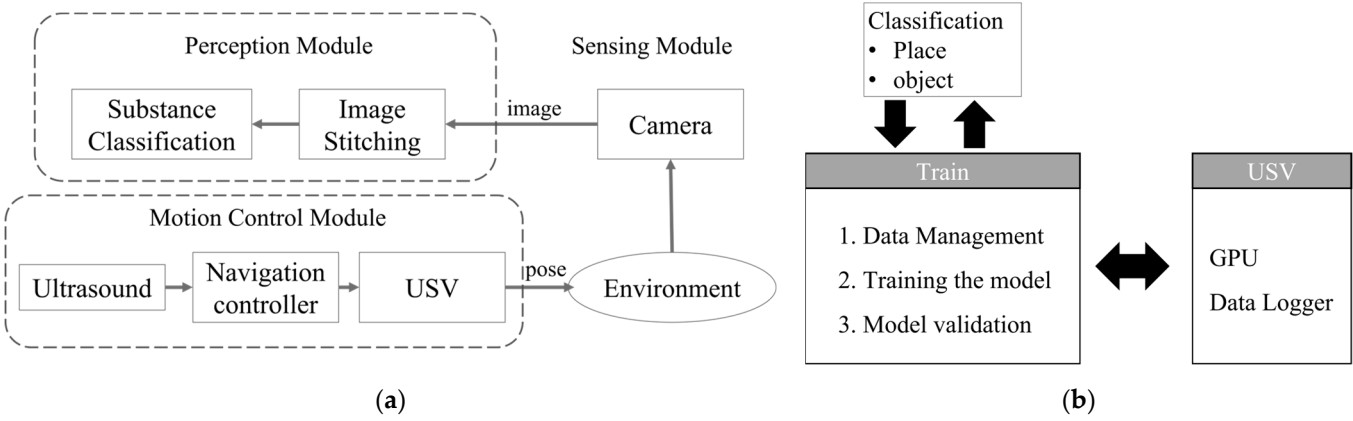

(**a**)                                                              (**b**)

**Figure 4.** Overall system diagram as (**a**) system module; (**b**) system data training and classification.

2.2.1. Feature-Based Panoramic Image Stitching

This system uses feature point detection as the benchmark for splicing. Feature point detection refers to the method of finding the feature points in the image based on the brightness, color, gradient, and other information of the image. In image alignment, feature point detection can be used to obtain feature points of two images, and then the alignment can be completed by matching these feature points. Common feature point detection methods such as: Harris Corner Detection, and Scale Invariant Feature Transformation (SIFT) [28–30]. As well, this system adopts SIFT method to carry on the detection, the flow chart is shown in Figure 5.

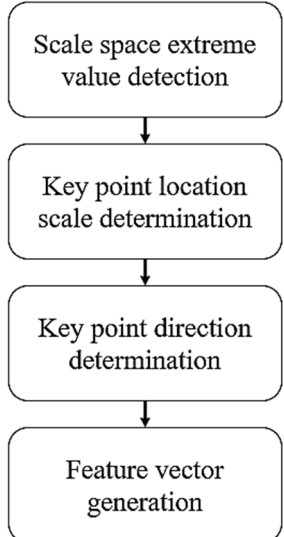

**Figure 5.** SIFT feature vector generation steps.

In the scale space extreme value detection, the Gaussian convolution kernel is applied due to its scale invariance. It detects the key points in the SIFT algorithm. The images are convolved using Gaussian filters at different scales, and then continuous Gaussian blurring of the image differences is used to find the key points. The key point is based on the maximum and minimum values of difference of Gaussians (DoG) at different scales:

$$L(x, y, \sigma) = G(x, y, \sigma) * I(x, y), \tag{1}$$

$$G(x, y, \sigma) = \frac{1}{2\pi\sigma^2} e^{-(x^2+y^2)/2\sigma^2}, \tag{2}$$

$$D(x, y, \sigma) = L(x, y, k_i\sigma) - L(x, y, k_j\sigma), \tag{3}$$

where $L(x, y, \sigma)$ is the image of the original image $I(x, y)$ convoluted with Gaussian mask $G(x, y, \sigma)$, $L(x, y, \sigma)$ is the DoG image. The maximum and minimum values in the DoG image are defined as key points as

$$m(x, y) = \sqrt{(L(x+1, y) - L(x-1, y))^2 + (L(x, y+1) - L(x, y-1))^2}, \tag{4}$$

$$\theta(x, y) = \tan^{-1} \frac{L(x, y-1) - L(x, y+1)}{L(x-1, y) - L(x+1, y)}, \tag{5}$$

where the magnitude and orientation of the key point are $m(x, y)$ and $\theta(x, y)$ respectively. Each adjacent pixel is added to the histogram of the key point according to its magnitude and direction, and the direction of the maximum value in the final histogram is the direction of the key point. Each extracted point will have three pieces of information: scale, coordinates, and direction. To improve the stability of the registration of selected points, each point is represented by $4 \times 4$, a total of 16 seed points, each point contains 128 data,

and the SIFT feature vector represented by the result is 128 dimensions. In this way, an image feature descriptor is generated for image feature matching, and the SIFT feature vector is no longer interfered by changes in direction and angle.

With these feature vectors, it is necessary to perform subsequent key point matching on the key points. The identification of the target is completed by the comparison of key point descriptors in the two-point set. The similarity index of the key point descriptor with 128 dimensions is used as follows:

$$R_i = (r_{i1}, r_{i2}, \ldots, r_{i128}),$$ (6)

$$S_i = (s_{i1}, s_{i2}, \ldots, s_{i128}),$$ (7)

$$d(R_i, S_i) = \sqrt{\sum_j^{128} (r_{ij} - s_{ij})^2},$$ (8)

where $R_i$ is the key point descriptor in the reference image, $S_i$ is the key point descriptor in the observation image, and $d(R_i, S_i)$ is the two similarity measures of the arbitrary reference image and observation image. The key point matching can be conducted by exhaustive method, but it will take too much time, so the data structure of the K-dimensional tree (K-d tree) is used to complete the search instead [31,32]. K-d tree is a binary tree in which each leaf node is a k-dimensional point. All non-leaf nodes can divide the space into two half spaces as a hyperplane. The content of the search is based on the key points of the target image, and the original image feature points that are closest to the feature points of the target image and the second adjacent original image feature points are searched.

### 2.2.2. Siamese-Based Target Tracking

Target tracking is to analyze the image sequence to calculate the position of the moving target in each frame of the image. Then, correlation matching is performed according to the characteristic values related to the moving target to obtain the complete trajectory of the target. The system uses a single target tracking method. The single target tracking (STT) method can predict the size and position of the target in subsequent frames. The basic task flow is shown in Figure 6 [33,34].

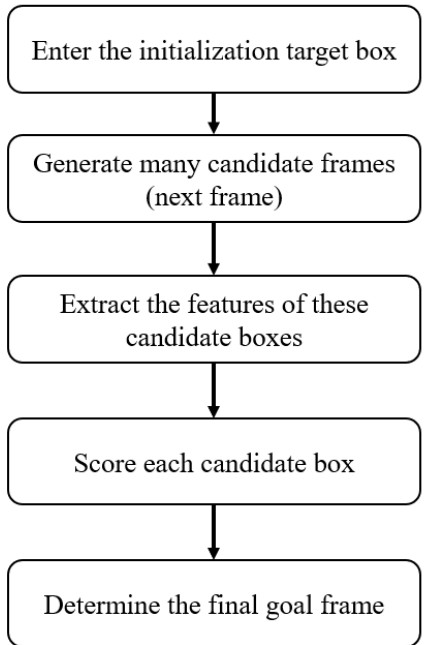

**Figure 6.** Basic structure and steps of STT system.

According to the above process, the first frame is selected in the movie, and then a lot of to-be-selected frames are generated in the next frame, and the feature values of these frames are extracted and scored. To adapt to changes in the appearance of the target and prevent drift in the tracking process, it is necessary to update the model approximately every frame. However, the past performance of the target is still important to the tracker. If it is continuously updated, it may lose the appearance information of the past performance and introduce too much noise. A combination of long and short-term updates can solve this problem. Therefore, the three steps of forgetting, updating, and output are added to the neural unit. The method of selecting the prediction results is generally divided into two categories: selecting the best one among multiple prediction results, using all the predicted values to weight the average, and then selecting.

Since the framed object may not be the relevant training set, it is impossible to track each frame of detection. Therefore, this paper needs to use deep learning to solve the above STT problem. This paper uses the Siamese region proposal network (SiamRPN) method to solve [35–37]. It can use large-scale images for offline end-to-end training. In general, this structure includes a Siamese subnetwork for feature extraction and a region proposal subnetwork. The candidate region generation network includes classification and regression. This network architecture is shown in Figure 7.

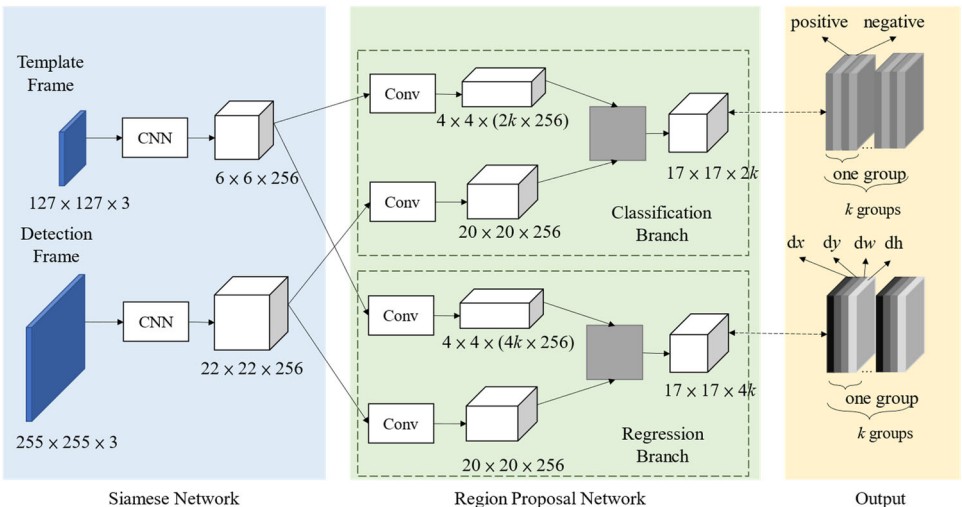

**Figure 7.** The main neural network framework of SiamRPN: the blue part is the Siamese subnet used for feature extraction, the green part is the regional proposal subnet, and the orange part is the final output result. There are two branches in the figure, one for classification and the other for regression.

First, SiamRPN uses multi-scale testing to predict the change of scale to solve the problem that the previous algorithm cannot estimate the size of the target. Because the sliding window method is time-consuming, the system uses RPN to directly generate the detection frame, which can greatly increase the generation speed. In addition, the anchor technology is used to determine whether there is a recognized target in the fixed reference frame, and how far the target frame deviates from the reference frame, so there is no need for multi-scale traversal of sliding windows. In the network architecture of Figure 7, the Siamese network uses the AlexNet network structures [38–40]. The Siam feature extraction subnet has two branches, and the two branches share the same parameters in the CNN. One is called the template branch, which receives the target patch as input in the previous frame. The other is called the detection branch, which receives the target in the current frame as input. $\varphi(z)$ is expressed as the output feature map of the template branch in the twin sub-network $6 \times 6 \times 256$. $\varphi(x)$ is expressed as the output feature map $22 \times 22 \times 256$ of the detection branches in the Siamese network. The connection network structure can be found that the template image and the search image are respectively obtained by the Siamese network with $6 \times 6 \times 256$ and $22 \times 22 \times 256$ features, and then

the template image features are respectively generated by a 3 × 3 convolution kernel features 4 × 4 × (2*k* × 256) and 4 × 4 × (4*k* × 256). In particular, the feature channel has been increased from the original 256 to 2*k* × 256 and 4*k* × 256. The reason why the number of channels has increased by 2*k* times is that *k* anchors are generated at each point of the feature map, and each anchor can be classified into the foreground or the background, so the classification branch has increased by 2*k* times. Similarly, each anchor can use four parameters as described, so the regression branch has increased by 4*k* times. At the same time, the search image also obtains two features through a 3 × 3 convolution kernel, where the number of feature channels remains unchanged. The RPN network architecture is shown in Figure 8, and the anchor method diagram is shown in Figure 9.

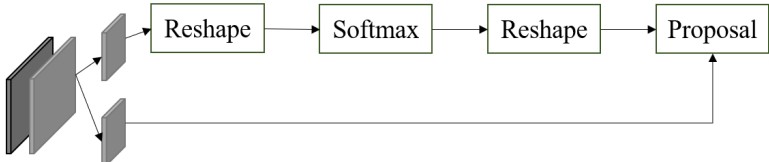

**Figure 8.** The RPN architecture used in this SiamRPN.

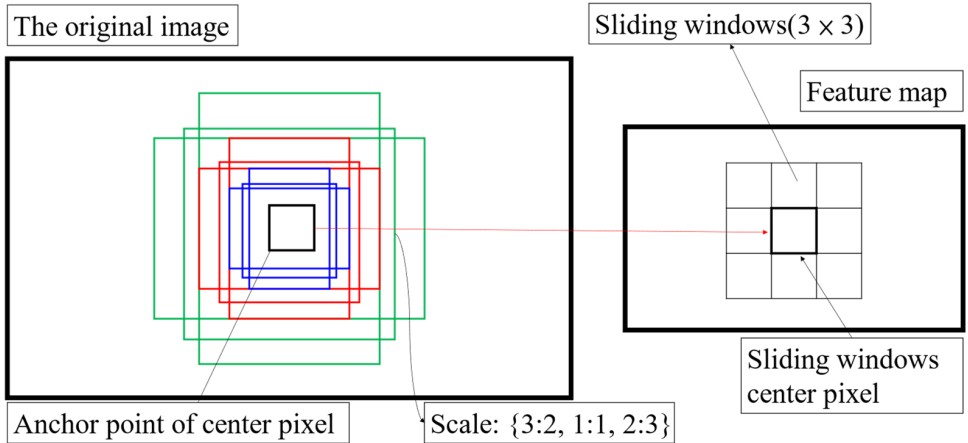

**Figure 9.** The anchor method architecture used in this SiamRPN.

The RPN network is divided into two lines [41] in comparison to the Kalman filter approach with a self-learning RBFNN (Radial Basis Function Neural Network) [42]. The upper one uses softmax classification to make the anchors obtain positive and negative classifications, and the lower one is used to calculate the bounding box regression offset for the anchors to obtain an accurate proposal. The final Proposal layer is responsible for synthesizing positive anchors and the corresponding bounding box regression offset to obtain proposals, and at the same time eliminate proposals that are too small and beyond the boundary. In Figure 9, there are nine rectangles in the anchor method and there are three shapes in total. The aspect ratio of the schematic diagram in Figure 9 is {1:1,2:3,3:2}, and the multi-scale method commonly used in detection is introduced. For the classification branch, the 4 × 4 × 256 features of the 2*k* template image anchors are used as the convolution kernel and the search image feature is convolved to generate the classification branch response map 17 × 17 × 2*k*. The same is used in the regression branch, and the response map generated after the convolution operation is 17 × 17 × 4*k*. Each point represents a vector of size 4*k*, which is d*x*, d*y*, d*w*, and dh. The deviation between the anchor and ground truth is measured, and the calculation formula of the response graph is as follows:

$$A^{cls}_{w \times h \times 2k} = [\varphi(x)]_{cls} \bullet [\varphi(z)]_{cls}, \tag{9}$$

$$A^{reg}_{w \times h \times 2k} = [\varphi(x)]_{reg} \bullet [\varphi(z)]_{reg}, \tag{10}$$

where $[\varphi(z)]_{cls}$ and $[\varphi(z)]_{reg}$ means that $\varphi(x)$ adds $2k$ classification channels and $4k$ regression channels, respectively, and • represents the calculation of the correlation on the classification branch and the regression branch. $A^{cls}$ contains $2k$ channel vectors, each point in it represents positive and negative excitation, which is classified by softmax loss. $A^{reg}$ contains $4k$ channel vectors, each point represents the dx, dy, dw, and dh between anchor and ground truth. The above is normalized by the Smooth L1 loss function as follows:

$$L_{reg} = \sum_{i=0}^{3} smooth_{L1}(\delta[i], \sigma), \tag{11}$$

$$\delta[0] = \frac{T_x - A_x}{A_w}, \delta[1] = \frac{T_y - A_y}{A_h}$$
$$\delta[2] = \ln\frac{T_w}{A_w}, \delta[3] = \ln\frac{T_h}{A_h} \tag{12}$$

$$smooth_{L1}(x, \sigma) = \begin{cases} 0.5\sigma^2 x^2, x \leq \frac{1}{\sigma^2} \\ x - \frac{1}{2\sigma^2}, x \geq \frac{1}{\sigma^2} \end{cases}, \tag{13}$$

where $A_x$, $A_y$, $A_w$, $A_h$ are the center point coordinates, length, and width of the anchor boxes; $T_x$, $T_y$, $T_w$, and $T_h$ are ground truth boxes. $L_{reg}$ is the final regression loss, $\delta$ is the coordinate standardization of the anchor, and $smooth_{L1}(x, \sigma)$ is the Smooth L1 loss function.

Algorithm 1 is a key frame marking method, which can be used in the detection and monitoring process of target tracking. In this paper, the tracking task is planned as a one-shot detection task. This method limits the input structure and automatically discovers features that can be generalized from new samples. That is to learn a learner net, which corresponds to the similarity function in this paper, is trained through a supervised Siamese network-based metric learning, and then reuses the features extracted by that network for one-shot learning. This research regards the target tracking task as a combination of one-shot object detection and few-shot instance classification. The former is a class-level subtask used to find candidate frames similar to the target, and the latter is an instance-level task used to distinguish targets and distractors. Target-guidance module distinguishes the characteristics of the target and search area and their interaction with the subject. Although the detector is focused on objects related to the target, the surrounding background interference is ignored. To compensate for this, a few-shot instance classifier is proposed. However, training directly from scratch is time-consuming and easily leads to overfitting. Therefore, few-shot finetune is performed through model-agnostic meta-learning, which enhances discrimination and further eliminates distractors. SiamRPN elaborated on it from a theoretical point of view, in which the tracking framework is shown in Figure 10.

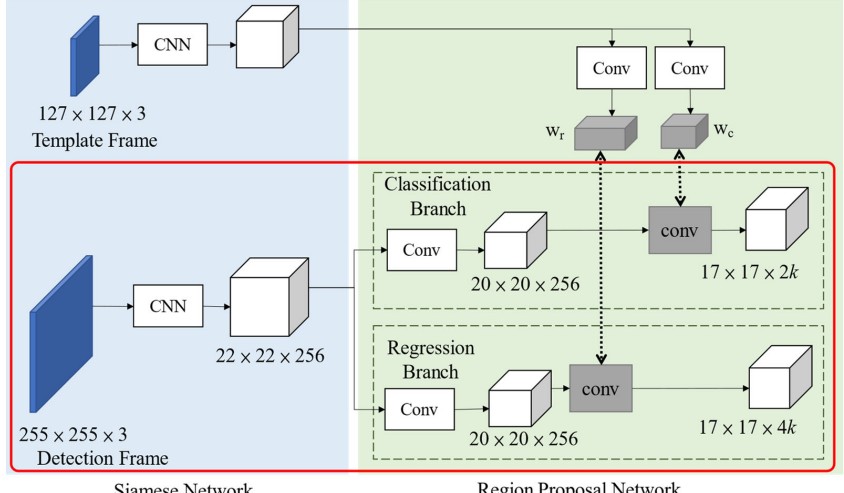

**Figure 10.** The tracking framework in SiamRPN.

| **Algorithm 1** Keyframe detection | | |
| --- | --- | --- |
| **Inputs:** | | $f$, the frame of the input video stream; $MAM_f$, the motion appearance mask of $f$; $MAM_{f-1}$, the motion appearance mask of $f-1$; $t_{stop}$, the temporal threshold for detecting stop; $|MAM_f|$ denotes the total number of ls in $MAM_f$; |
| **Outputs:** | | $l_f$, the label of the keyframe; |
| | 1. | if $|MAM_f| > |MAM_{f-1}|$ |
| | 2. | $l_f$ = SPLIT |
| | 3. | else if $|MAM_f| < |MAM_{f-1}|$ |
| | 4. | $l_f$ = JOIN |
| | 5. | else if $MAM_f {\,\hat{}\,} MAM_{f-1} \neq 0$ |
| | 6. | $l_f$ = MOVE |
| | 7. | else /*$MAM_f = MAM_{f-1}$ */ |
| | 8. | stop-count $\leftarrow$ stop-count + 1 |
| | 9. | if stop-count > $t_{stop}$ |
| | 10. | $l_f$ = SPLIT |

In the detection part, according to this network architecture, the single sample detection task can be shown as the red box. The initial frame of the template frame passes through the convolutional layer in the RPN, and $\varphi(x)_{reg}$ and $\varphi(x)_{cls}$ are used. In the detection part, the average loss function, the definition of the classification feature map and the regression feature map formula are as follows:

$$\min_W \frac{1}{n} \sum_{i=1}^{n} L(\zeta(\varphi(x_i; W); (\varphi(z_i; W)), l_i), \tag{14}$$

$$A^{cls}_{w \times h \times 2k} = \left\{ (x_i^{cls}, y_j^{cls}, c_l^{cls}) \right\}, \tag{15}$$

$$A^{reg}_{w \times h \times 4k} = \left\{ (x_i^{reg}, y_j^{reg}, dx_p^{reg}, dy_p^{reg}, dw_p^{reg}, dh_p^{reg}) \right\}, \tag{16}$$

where (14) is the average loss function $L$, $l_i$ is the label, $W$ is the weight of two networks, and $\zeta$ is the RPN operation. In (15), where $i \in [0, w)$, $j \in [0, h)$, $l \in [0, 2k)$. In (16), where $i \in [0, w)$, $j \in [0, h)$, $p \in [0, k)$. With the defined equation, the following relational equation, and the basis for selecting the best frame can be calculated:

$$CLS^* = \{ (x_i^{cls}, y_j^{cls}, c_l^{cls})_{i \in I, j \in J, l \in L} \}, \tag{17}$$

$$ANC^* = \{ (x_i^{an}, y_j^{an}, w_l^{an}, h_l^{an})_{i \in I, j \in J, l \in L} \}, \tag{18}$$

$$REG^* = \{ (x_i^{reg}, y_j^{reg}, dx_l^{reg}, dy_l^{reg}, dw_l^{reg}, dh_l^{reg})_{i \in I, j \in J, l \in L} \}, \tag{19}$$

$$PRO^* = \{ (x_i^{pro}, y_j^{pro}, w_l^{pro}, h_l^{pro})_{i \in I, j \in J, l \in L} \}, \tag{20}$$

$$x^{pro} = x^{an} + dx_l^{reg} * w_l^{an}, \tag{21}$$

$$y^{pro} = y^{an} + dy_l^{reg} * h_l^{an}, \tag{22}$$

$$w_l^{pro} = w_l^{pro} * e^{dw_l}, \tag{23}$$

$$h_l^{pro} = h_l^{pro} * e^{dh_l}, \tag{24}$$

where the top $k$ values in the positive score found in the classification score are *CLS\**. It is found that the anchor box of the corresponding box is found to be *ANC\** and the predicted regression value is *REG\**, and finally the regression value is converted to the regression box *PRO\**. In the system, $^{an}$ represents the orbit generated by the anchor, and $^{pro}$ is the bounding

box that is finally returned. Then, the anchors that are too far from the center are discarded to remove outliers, and then non-maximum suppression (NMS) is used to remove all non-maximum frames to remove redundant overlapping frames. The intersection over union (*IoU*) and NMS formulas needed to select the best frame are as follows:

$$IoU = \frac{\text{Area of Intersection}}{\text{Area of Union}}, \quad (25)$$

$$(w + p) \times (h + p) = s^2, \quad (26)$$

where *IoU* is the union of the intersection ratio of the two box areas, which is used to determine the pixel distance of the two boxes. In (26), $w$ and $h$ represent the width and height of the target, and p represents the filling value equal to $(w + h)/2$. First, select a box with the highest credibility, the rest of the boxes, and their *IoU* are greater than a certain threshold, then remove them, continue to select a box with the highest score from the unprocessed box, and repeat the above process to obtain the best.

## 3. Results

The target tracking currently used on USVs rarely uses Siamese neural network architecture and most of them are single-hull structures. The overall effect of this research is to achieve system integration, which includes three main modes: (1) fixed-way cruise and 360-degree panoramic monitoring. (2) real-time target tracking and USV following. (3) in the range Internal launch feedback module. To achieve the above objectives, this section will mention all the specifications, circuit diagrams, and algorithm results adopted by the system. The final product diagram and specification of the overall system are shown in Figure 11.

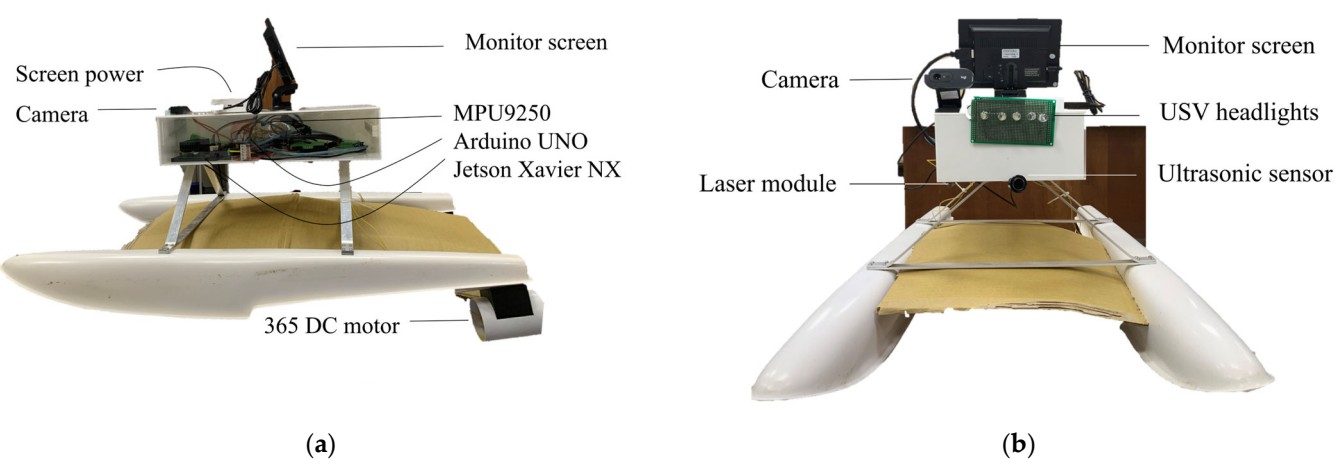

(**a**)        (**b**)

**Figure 11.** The USV as: (**a**) side view; (**b**) front view.

In order to estimate the difference between the USV's travel route and the preset route, as shown in Figure 12, this paper uses two different routes for comparison. Path one is an arc-shaped curve, in which the trajectory with a straight line at the beginning can be used to compare whether it can meet the preset value in a straight line. Path two is a route to avoid obstacles and currents and is used to judge whether the USV can sail as usual under different wind speeds. Among them, each path has four tests at different times. The record of different trajectories path one and path two is shown in Figure 13. The green path is the first test, the purple path is the second test, the light blue path is the third test, and the dark blue path is the fourth test. In the four-day test, there was no wind and no water flow on the first day, light breeze on the second day, weak water flow, strong wind, and strong water flow on the third day, and light breeze but strong water flow on the fourth day.

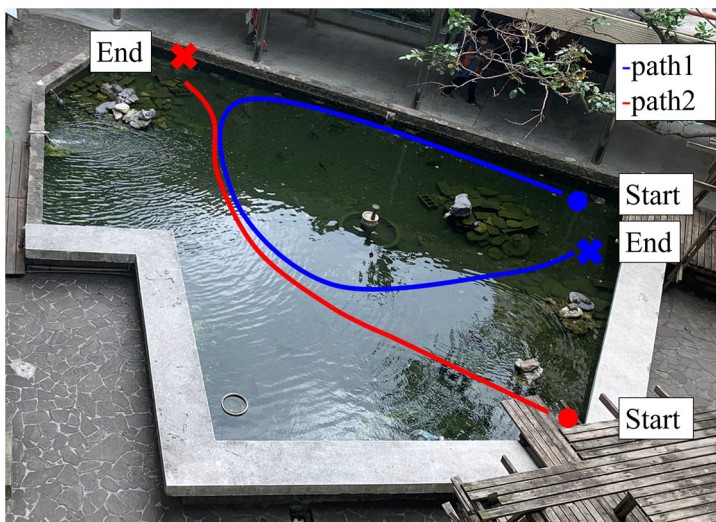

**Figure 12.** Two paths tested in established waters.

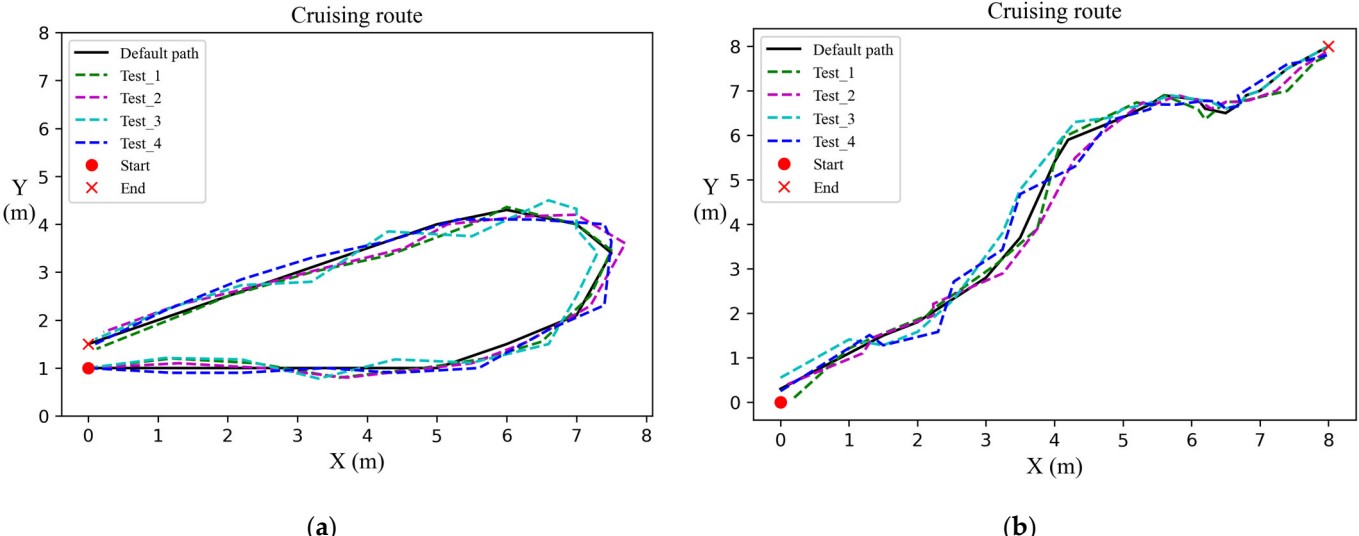

**Figure 13.** T Four test routes and preset routes, listed as: (**a**) path 1; (**b**) path 2.

The route information obtained from the four tests on path one and path two will be evaluated and verified with absolute trajectory error (ATE), as shown in Table 1. It can be seen from this that although the catamaran has strong resistance to water currents, it still has some continuous drop compared to static waters (path two). In the turning part, the two routes will deviate slightly, which may be caused by the wind speed change at that time or other underwater biological activities. It can be seen that the USV can fit as accurately as possible under no wind or breeze, and it can be overcome even under strong water flow. However, the performance is not satisfactory when encountering strong winds. The second part of the algorithm part is a panoramic stitching method based on feature comparison. The general method is to capture the feature points of two photos, and then connect the corresponding points together. The following paper will show the detailed stitching process step by step. This paper compares two pictures, and finally will provide the final complete stitching picture. All the steps and derivative pictures of the splicing method are shown in Figures 14–18.

**Table 1.** Detailed specifications of the feedback module.

| Trajectory | Path1 | Path2 |
|---|---|---|
| Metric | ATE | |
| Test_1 | 3.745 | 4.045 |
| Test_2 | 3.749 | 3.704 |
| Test_3 | 7.341 | 19.363 |
| Test_4 | 6.958 | 4.499 |
| Avg. Err. | 5.473 | 7.903 |

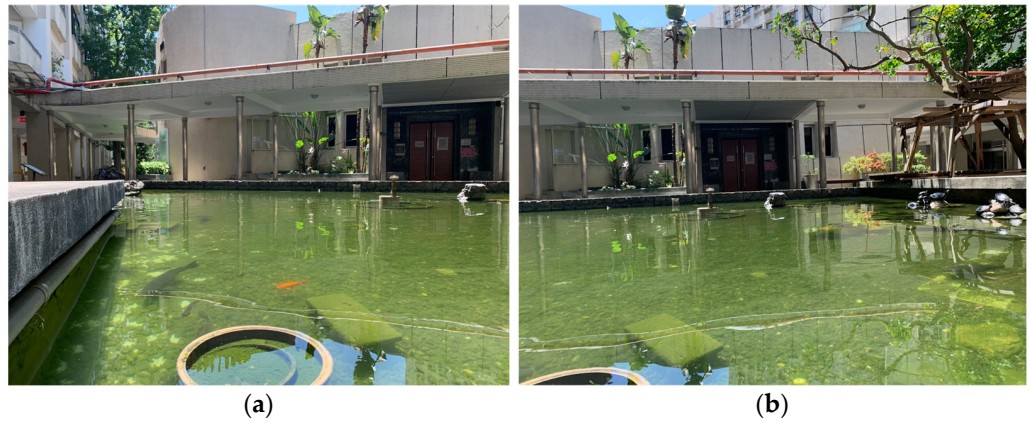

(**a**)        (**b**)

**Figure 14.** The original picture of feature-based image alignment. (**a**) Picture to the left after centering the camera. (**b**) Picture to the right after centering the camera.

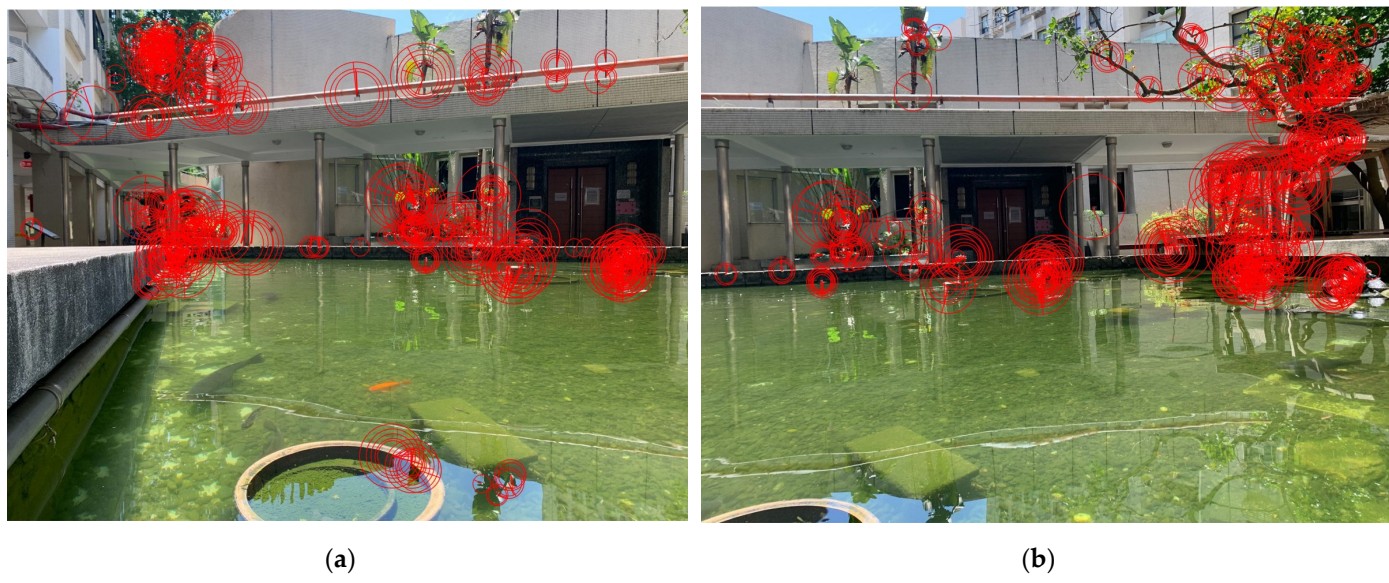

(**a**)        (**b**)

**Figure 15.** Key points with the descriptors detected as in (**a**) Figure 14a. (**b**) Figure 14b.

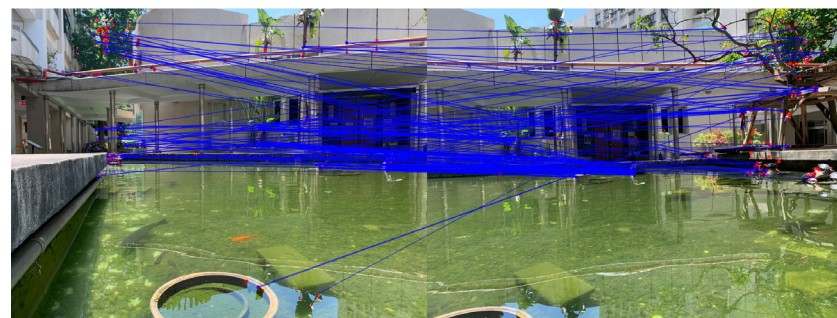

**Figure 16.** Feature points matched.

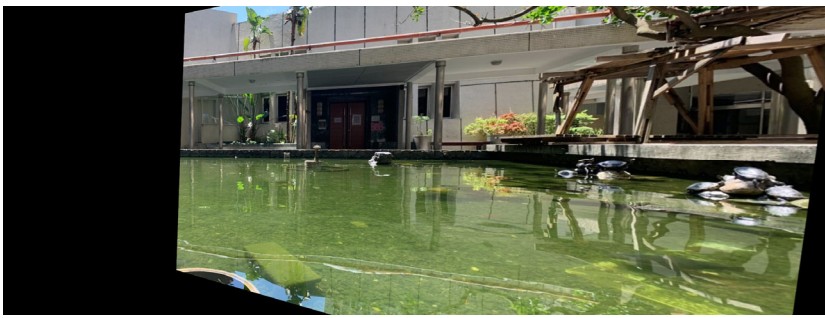

**Figure 17.** Homograph corresponding to the two photos, Figure 14b is rotated and deformed based on Figure 14a.

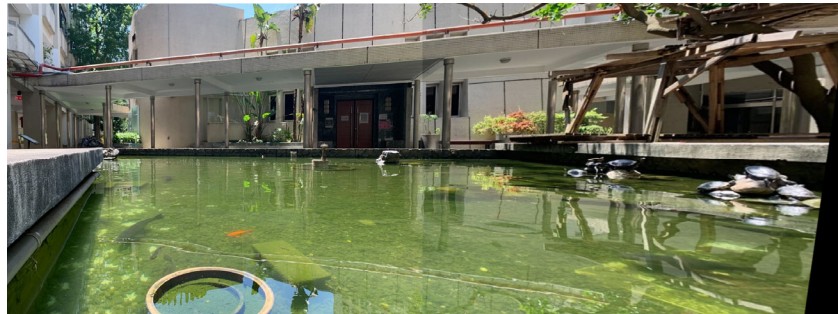

**Figure 18.** Final corrected wide-angle image.

The first step is to find and filter the key points and corresponding points in the photo. One can see the red circle in Figure 15. A key point will have four circles. The larger the circle, the more obvious the feature. The second step is to perform feature matching. There are many matching blue lines in Figure 16. Among them, this study only uses the feature points that have the distribution ratio reaching the top 85% score. The third step is Homography, which is a reversible transformation from the real projective plane to the projective plane. The straight line is still mapped to a straight line under this transformation, that is to say, the method of expanding the 3D plane to the plane. In the end, it is to merge the pictures to achieve the goal. The picture obtained in actual navigation according to the feature method is shown in Figure 17, and the picture obtained by synchronous stitching in actual navigation is shown in Figure 18.

The wide-angle image obtained is shown in Figure 19. This paper takes up to 500 key points and only selects the top 15% points for matchmaking.

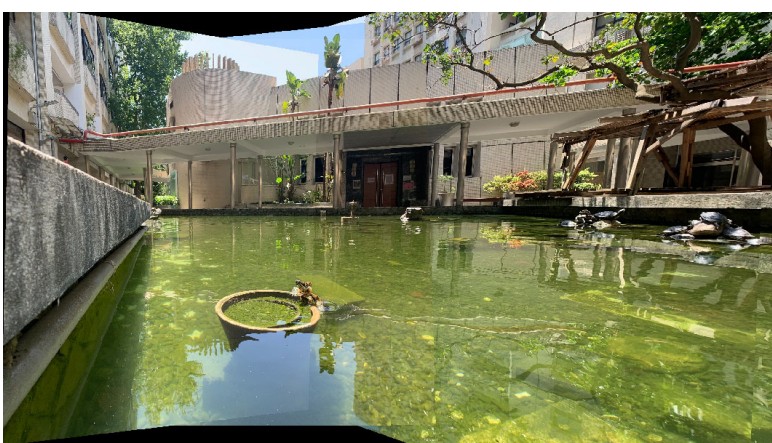

**Figure 19.** The picture obtained by synchronous stitching in actual navigation according to the feature method.

The third part of the algorithm part is target tracking. This paper compares and analyzes SiamRPN, efficient convolution operators for tracking (ECO), continuous convolution operators (C-COT), and DaiSiamRPN target trackers, and proves the reliability of using SiamRPN. In addition, this paper will use the visual object tracking 2018 (VOT2018) and object tracking benchmark 100 (OTB100) data sets for short-term target tracking comparison. The evaluation index comparison in this section is used to discuss which tracker method to choose. VOT2018 is divided into four items: accuracy, robustness, loss, and expected average overlap (EAO) as the basis for verification and evaluation. The detailed calculation standards are as follows:

$$\Phi_t(i) = \frac{1}{N_{rep}} \sum_{k=1}^{N_{rep}} \Phi_t(i,k), \tag{27}$$

$$\rho_A(i) = \frac{1}{N_{valid}} \sum_{j=1}^{N_{valid}} \Phi_j(i), \tag{28}$$

$$\rho_R(i) = \frac{1}{N_{rep}} \sum_{k=1}^{N_{rep}} F(i,k), \tag{29}$$

$$\hat{\Phi} = \frac{1}{N_{hi} - N_{1o}} \sum_{N_s = N_{1o}:N_{hi}} \hat{\Phi}_{N_S}, \tag{30}$$

$$\text{Overlap score } (OS) = \frac{|a \cap b|}{|a \cup b|}, \tag{31}$$

where the accuracy rate refers to the average overlap rate of the tracker in the test, that is, the *IoU* algorithm is used to compare the overlapping area of two rectangular boxes divided by the total area of the two rectangular boxes. $\Phi_t(i)$ is the definition of the average accuracy of each frame, where $\Phi_t(i,k)$ represents the accuracy of the $i$th tracking in repeated $k$ frames. $\rho_A(i)$ is the average accuracy of the entire video. Robustness refers to the number of failures of the test tracker. When the overlap rate of the rectangular frame is zero, it is judged as a failure, so the higher the value, the better. In (29), the function $F(i,k)$ is defined as the number of tracking failures, and the measurement is repeated at the $k$th algorithm. EAO is the expected value of the non-reset overlap of each tracker on the short-term image sequence and is the most important indicator for evaluating the accuracy of the VOT target tracking algorithm. Success means that if the conformance rate score is higher than a certain value, it is regarded as a success. The higher the value, the better. $\Phi_{N_s}$ is the average coverage of the $Ns$ in the video, and $\Phi(i)$ is the accuracy between the predicted frame and the real frame. As the video frame increases, the average coverage value will decrease because $\Phi(i) \leq 1$. In (31), the bounding box obtained by the tracking algorithm is $a$, and

the box given by the ground-truth is $b$. When the OS of a frame is greater than the set threshold, the frame is regarded as Success, and the percentage of all successful frames to all frames is the success rate. The values of VOT2018 and OTB100 in each tracker are shown in Tables 2 and 3, SiamRPN, ECO, C-COT, and DaiSiamRPN are compared. Target tracking in actual navigation is shown in Figure 20.

**Table 2.** Evaluation of VOT2018 by the system.

| Tracker | Accuracy ($\rho_A(i)$) | Robustness ($\rho_R(i)$) | EAO ($\Phi$) |
|---|---|---|---|
| SiamRPN | 0.601 | 0.337 | 0.318 |
| ECO | 0.484 | 0.276 | 0.281 |
| C-COT | 0.536 | 0.184 | 0.378 |
| DaiSiamRPN | 0.601 | 0.337 | 0.327 |

**Table 3.** Evaluation of OTB100 by the system.

| Tracker | Success (OS) | Precision |
|---|---|---|
| SiamRPN | 0.694 | 0.914 |
| ECO | 0.691 | 0.910 |
| C-COT | 0.671 | 0.898 |
| DaiSiamRPN | 0.658 | 0.881 |

In Figure 20, it can be seen that good results can be achieved in all stages of target tracking, which improves the previous SiamFC problem. The SiamRPN selected in the system is the best tracker under comprehensive comparison. The AlexNet training loss as the backbone is shown in Figure 21. The blue line represents training, and the red line represents validation.

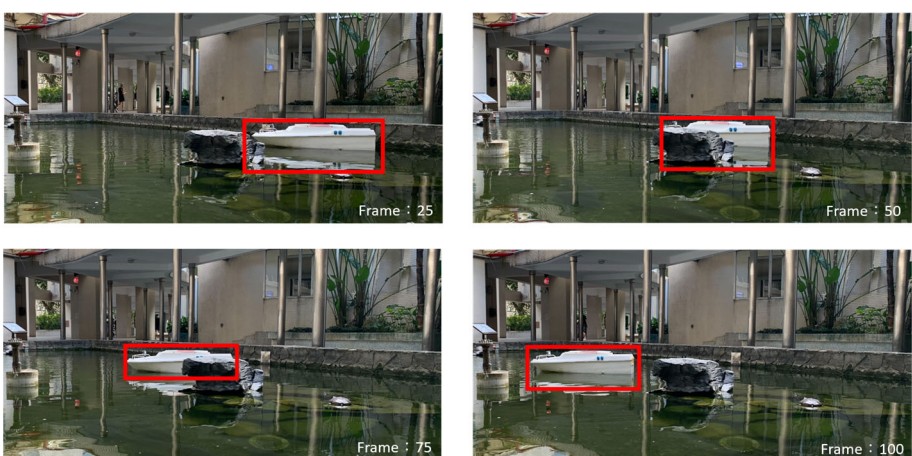

**Figure 20.** Target tracking of different scales in actual system navigation.

This study collected two different data sets: daytime and night. In these data sets, two frames per second are used as tests, and several ships of different durations are used for recording. Figure 22 is a database of different USV data during daytime and night. In the daytime and nighttime data, it can be found that the red hull target tracking efficiency is higher in the daytime data set. It may be because when the sunlight is too strong and the hull is brighter, which will cause reflections, making the lens unable to obtain sufficient feature values. In the night data set, the tracking efficiency of the white hull target is higher. The USVs have searchlights that illuminated on the tracked object in low light conditions, and the brighter the hull, the higher the feature variance can be obtained in contrast to the background. The following will test and verify the daytime data set and night data set,

respectively. Among them, the precision, recall, and F1 scores of the day data set and night data set are shown in Table 4 below.

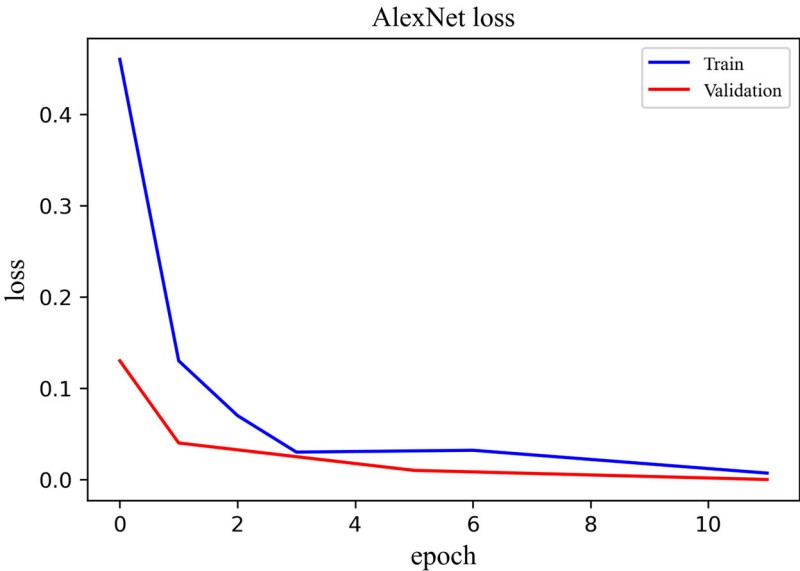

**Figure 21.** Target tracking of different scales in actual system navigation.

**Table 4.** Daytime and night data set evaluation.

| Data Set | Precision | Recall | F1 Scores |
|----------|-----------|--------|-----------|
| Daytime | 0.85 | 0.61 | 0.711 |
| Night | 0.74 | 0.82 | 0.778 |

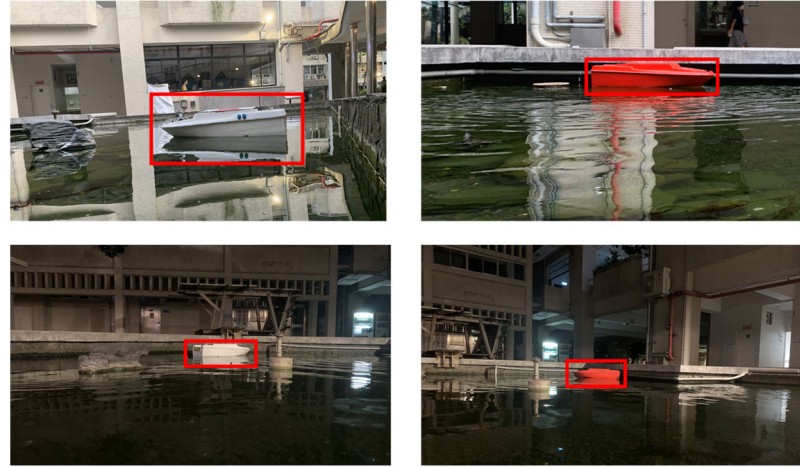

**Figure 22.** Different USV data sets during daytime and night.

The P-R curve can be obtained after the sum of the information obtained from the data set as shown in Figure 23. The confusion matrix of the daytime data set and the nighttime confusion matrix are shown in Tables 5 and 6, respectively. The receiver operating characteristic (ROC) curve obtained through the sum of the two confusion matrices is shown in Figure 24.

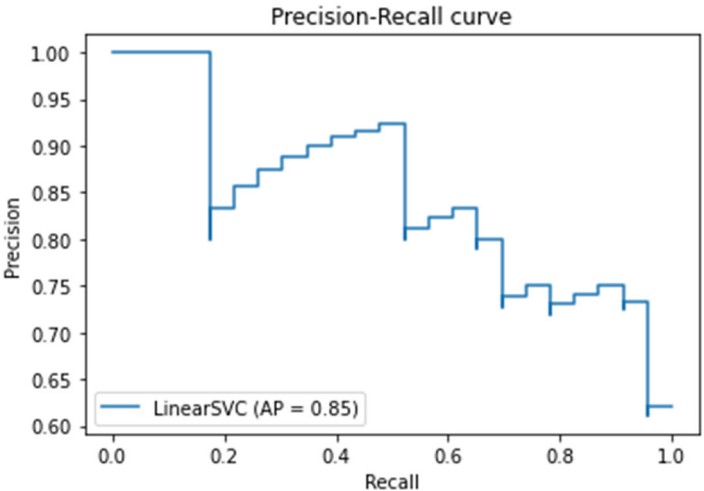

**Figure 23.** P-R curve of the sum of information obtained by the data set.

**Table 5.** Confusion matrix of daytime in SiamRPN.

| Day-Time Data Set | | Predicted | |
|---|---|---|---|
| | | Positive | Negative |
| Actual | Positive | 1250 | 782 |
| | Negative | 221 | 575 |

**Table 6.** Confusion matrix of night in SiamRPN.

| Night-Time Data Set | | Predicted | |
|---|---|---|---|
| | | Positive | Negative |
| Actual | Positive | 751 | 166 |
| | Negative | 263 | 323 |

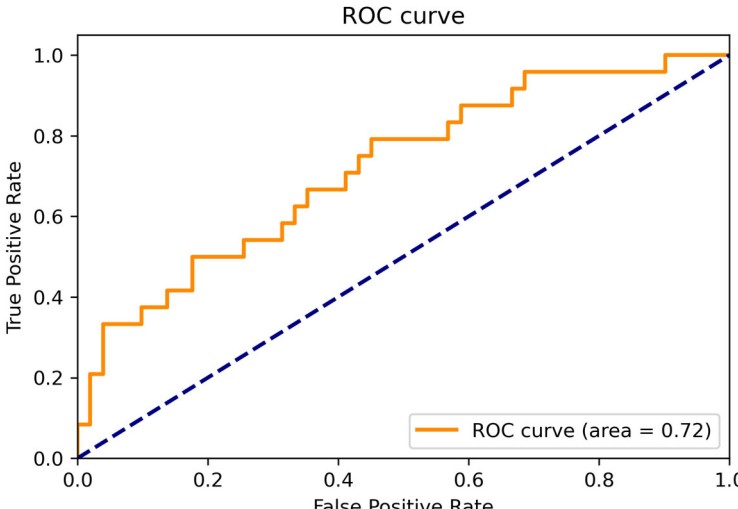

**Figure 24.** ROC curve of the data set.

According to the above result evaluation data, precision and recall can achieve good results when the data sets are mixed. The area under the ROC curve is 0.72, which shows the superior performance of this system. The difference between light and dark can be clearly seen. In the daytime data set, the precision is higher, and the recall is lower, and in the night data set, the precision is lower, and the recall is higher. As far as the hardware is concerned,

this study has verified the stability and proved that the stability of the catamaran is better than that of the monohull. In terms of software, SiamRPN, which has better accuracy and real-time performance, was chosen.

## 4. Discussion

This research has achieved the use of a USV to navigate, avoid obstacles, and perform target tracking, while achieving precise navigation, high accuracy of algorithm tracking, and high vehicle following speed. It can be clearly seen from the results that the system has significant improvements and enhancements to the above parts. In this section, it will be divided into hardware, algorithm, and integration parts for detailed discussion.

In the hardware part, this paper uses the information of the three-axis accelerometer and the three-axis gyroscope to compensate for the sensor in the system, which can improve the reliability of the sensor on the wave surface. In Figure 12, the Z-axis movement of the path before the compensation is too abrupt in the rotation stage, and the Z-axis movement of the path after the compensation is relatively smooth in the rotation stage. In addition, it replaced the monohull with Catamaran to improve stability and speed. Catamaran is that the waterline area is small, the interference force of the waves is small, and it has superior resistance in waves. In Figure 13, after the same use of the compensator, the monohull will amplify the wave amplitude due to its own hardware shortcomings, causing the information obtained by the sensor to be too extreme and lack reliability.

In the algorithm part, first, the system uses feature-based image stitching to expand the original viewing angle of only 55 degrees to a wider area without missing important information. In target tracking, this algorithm uses RPN so that it can instantly change the circle frame and track accurately, which is the best method compared to the comprehensive performance of ECO, C-COT, and DaSiamRPN. In the comparison between VOT2018 and VOT2018-LT, the effectiveness of SiamRPN and DaSiamRPN in Accuracy, Robustness, and EAO is significantly higher than that of ECO and C-COT. For the indicators in OTB: Success and Precision, SiamRPN performs better than DaSiamRPN. In addition, the effect of this algorithm on the test set is excellent, and the AOU can reach 0.72.

In the integration part, the most challenging is to make immediate responses to the sensing components and feedback components. It was originally expected that all sensing components and feedback components were placed on Jetson Xavier NX, but in experiments, it was found that if the target tracking algorithm and motor were activated, the remaining GPIO pins could not provide enough current to drive. In this study, methods such as pull-up resistors have been tried, and in the end, the use of dual control boards to interact with each other was chosen to achieve the best efficiency. This system uses Arduino and Jetson Xavier NX to communicate via USB using Python. The sensing component obtains the value through Arduino and sends it back to the main control board Jetson Xavier NX for judgment, correction, compensation, and response.

Other than our approach, various different approaches have been proposed as the use of laser scanners as machine vision systems in Unmanned Aerial Vehicle (UAV) navigation when compared with camera-based systems [43], autonomous robotic group behavior optimization during the mission on a distributed area in a cluttered hazardous terrain [44] and the machine vision systems to determine physical values of near distanced objects for Unmanned Aerial Vehicle (UAV) navigation [45].

## 5. Conclusions

In this research, it is proposed to use SiamRPN as USV target tracking and IMU as feedback to accurately locate ships and navigate fixed routes. This research is achieved under a special USV and embedded system. Because the USV is lighter and faster, it can be applied to the pursuit and rescue of smugglers. The main contributions of this paper are: (1) Improve the slow and poor accuracy of target tracking on common vehicles. (2) Combine IMU for dual-hull vehicles to improve deviated trajectories and wave undulations. (3) Combining image stitching methods based on feature points to reduce blind angles of

sight. The experimental results show that this research can achieve target tracking and automatic navigation in different waters. Among the scheduled routes, this paper uses Catamaran's way to replace monohulls to improve stability and speed. The method of image stitching is used to improve the problem of the blind angle of the viewing angle, so that the USV will not lose important information. At present, due to the current problem of the embedded system, two control boards are needed to meet all the requirements. In the future, it can be towards adding multiple lenses or 360-degree lenses to reduce the burden of algorithms and reduce other feedback sensors to reduce the current burden of the control board. For a future work, (1) more experiments on system stability of USV should be conducted over various water surface environments, (2) arbitrary multiple obstacles experiments should be carried out, and (3) more specifications should be evaluated (e.g., the duration to performs the tasks of object recognition and tracking on the water surface.), (4) comparison between monohull and catamaran type vehicle and (5) experiment and implement other control techniques to compare with the presented results.

**Author Contributions:** Conceptualization, M.-F.R.L.; methodology, M.-F.R.L. and C.-Y.L.; software, C.-Y.L.; validation, M.-F.R.L. and C.-Y.L.; formal analysis, M.-F.R.L.; investigation, M.-F.R.L.; resources, M.-F.R.L.; data curation, C.-Y.L.; writing—original draft preparation, C.-Y.L.; writing—review and editing, M.-F.R.L.; visualization, C.-Y.L.; supervision, M.-F.R.L.; project administration, M.-F.R.L.; funding acquisition, M.-F.R.L. All authors have read and agreed to the published version of the manuscript.

**Funding:** This research was funded by [Ministry of Science and Technology (MOST) in Taiwan] grant number [108-2221-E-011-142-] and [Center for Cyber-physical System innovation from the Featured Areas Research Center Program within the framework of the Higher Education Sprout Project by the Ministry of Education (MOE) in Taiwan].

**Institutional Review Board Statement:** Not applicable.

**Informed Consent Statement:** Not applicable.

**Data Availability Statement:** Not applicable.

**Conflicts of Interest:** The authors declare no conflict of interest.

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
