# Peer review of "Object Tracking for an Autonomous Unmanned Surface Vehicle"

_machines, doi:10.3390/machines10050378_

Round 1

Reviewer 1 Report

The paper presents an interesting description and application of control strategies and obstacles recognition for an autonomous unmanned surface vessel. Some minor comments are listed below:

  1. The title of the paper is a little bit confusing, as the paper is not dealing with the design of the vessel, but with the design of the control system of the vessel only. If the title remains the same, dedicated sections should be provided for a proper description of the vessel itself (hull form, stability, manoeuvring characteristics, etc...), properly justifying the choice of a catamaran instead of a monohull.
  2. No comparison between monohull and catamaran is provided concerning dimensions, propulsion and steering system.
  3. It is not clear if the authors experimented and implemented other control techniques to compare with the presented results.
  4. In Figure 25, why the catamaran path is not starting from the initial fixed point but has an offset of about 0.5 meters in the Y direction?
  5. It is suggested to add also the obstacle position in the path figures.
  6. Figures 26 and 27 would be more readable in 2D form (XZ plane) to have a better comparison of the Z values.
  7.  Please improve the quality of Figure 45, to have the same quality as the other provided graphs in the paper.

Reviewer 2 Report

1) In Figure 2, the AI model is deployed remotely to control the USV via wireless communication. However, since the data packages of images are relatively large, and the wireless signal on the water is not stable enough. These situations can affect AI model to control the USV. Please explain how to solve these situations when the communication between local and remote is abnormal.

2) Please explain the meaning and implementation of formation keeping in the Figure 3.

3) Has USV conducted any experiments of system stability? Please explain how long USV performs the tasks of object recognition and tracking on the water surface.

4) Suggest that the authors carry out the experiment with arbitrary multiple obstacles, there just only one obstacle in this manuscript .

Reviewer 3 Report

This contribution presents original ideas in the study and advances the previous research in this area. The level of the originality of contribution to the existing knowledge with an emphasis on the paper’s innovativeness in both theory development and methodology used in the study is very high.

This work makes a significant practical contribution and it makes impact on the research work on the research community.

The quality of arguments, the critical analysis of concepts, theories and findings, and consistency and coherency of debate are well addressed in this paper. 

The paper has a good writing style in term of accuracy, clarity, readability, organization, and formatting. 

  • In Figure 2. an unmanned vehicle control architecture diagram is presented, please adress the problem of the stability of the control strategy.
  •  Can equation (29) present some singularity? Please discuss.

Concerning the cited literature you can consider the following papers to improve the tutorial aspects of the paper.

L. Lindner et al., "Machine vision system errors for unmanned aerial vehicle navigation," 2017 IEEE 26th International Symposium on Industrial Electronics (ISIE), 2017, pp. 1615-1620, doi: 10.1109/ISIE.2017.8001488.

M. Ivanov et al., "Influence of data clouds fusion from 3D real-time vision system on robotic group dead reckoning in unknown terrain," in IEEE/CAA Journal of Automatica Sinica, vol. 7, no. 2, pp. 368-385, March 2020, doi: 10.1109/JAS.2020.1003027.

L. Lindner et al., "Machine vision system for UAV navigation," 2016 International Conference on Electrical Systems for Aircraft, Railway, Ship Propulsion and Road Vehicles & International Transportation Electrification Conference (ESARS-ITEC), 2016, pp. 1-6, doi: 10.1109/ESARS-ITEC.2016.7841356.

Round 2

Reviewer 2 Report

The work maybe quite important, but it didn't present properly in this paper. The author addressed that the traditional ship sensing device is expensive and low accurate. But in this paper, there are tedious description of some basic algorithms and the information of the devices used, but not enough discussion about the accuracy. It reads more like a technical report other than a scientific article. Moreover, there are a lot of incomplete sentences in the paper. For example, line 53-line 62.
